# P-BiC: Ultra-High-Definition Image Moiré Patterns Removal via Patch Bilateral Compensation

Zeyu Xiao*
University of Science and Technology of China & National University of Singapore
Hefei, China & Singapore
zeyuxiao@mail.ustc.edu.cn

Zhihe Lu†
National University of Singapore
Singapore
zhihelu@nus.edu.sg

Xinchao Wang‡
National University of Singapore
Singapore
xinchao@nus.edu.sg

## Abstract

People nowadays use smartphones to capture photos from multimedia platforms. The presence of moiré patterns resulting from spectral aliasing can significantly degrade the visual quality of images, particularly in ultra-high-definition (UHD) images. However, existing demoiréing methods have mostly been designed for low-definition images, making them unsuitable for handling moiré patterns in UHD images due to their substantial memory requirements. In this paper, we propose a novel patch bilateral compensation network (P-BiC) for the demoiré pattern removal in UHD images, which is memory-efficient and prior-knowledge-based. Specifically, we divide the UHD images into small patches and perform patch-level demoiréing to maintain the low memory cost even for ultra-large image sizes. Moreover, a pivotal insight, namely that the green channel of an image remains relatively less affected by moiré patterns, while the tone information in moiré images is still well-retained despite color shifts, is directly harnessed for the purpose of bilateral compensation. The bilateral compensation is achieved by two key components in our P-BiC, *i.e.*, a green-guided detail transfer ($G^2DT$) module that complements distorted features with the intact content, and a style-aware tone adjustment (STA) module for the color adjustment. We quantitatively and qualitatively evaluate the effectiveness of P-BiC with extensive experiments. The code is publicly available at: https://github.com/zeyuxiao1997/P-BiC.

## CCS Concepts

• **Computing methodologies → Reconstruction**.

## Keywords

Image restoration, Image demoiréing, Ultra-high-definition image

---

*This work is done when Zeyu is a visiting student at the National University of Singapore, supported by Xinchao Wang.
†Zeyu and Zhihe contribute equally.
‡Corresponding author.

---

**ACM Reference Format:**
Zeyu Xiao, Zhihe Lu, and Xinchao Wang. 2024. P-BiC: Ultra-High-Definition Image Moiré Patterns Removal via Patch Bilateral Compensation. In *Proceedings of the 32nd ACM International Conference on Multimedia (MM '24), October 28-November 1, 2024, Melbourne, VIC, Australia.* ACM, New York, NY, USA, 10 pages. https://doi.org/10.1145/3664647.3681144

## 1 Introduction

Moiré patterns often arise when a camera captures a subject with a patterned texture or a repetitive structure that has a similar frequency to the camera's sensor, resulting in visual distortion and a loss of detail in the captured image. To eliminate moiré patterns and restore high-quality images, image demoiréing has been proposed and drawn increasing attention in both academia and industry. Recent works have made remarkable progress in improving the performance of image demoiréing by introducing advanced neural network architectures [6, 7, 14, 16, 19–21, 24, 26, 34, 39–42, 44, 46, 52, 53, 55]. However, those methods are primarily designed for removing the moiré patterns in low-definition (LD) images, often resulting in intense computational cost and poor performance due to more complex and pronounced moiré patterns when applied to high-definition (HD) images, *e.g.*, photographs acquired by present smartphones.

Recent works have been proposed to eliminate moiré patterns in HD and even ultra-HD (UHD) images [12, 18], which are becoming more popular in current imaging systems. Compared with LD images, the moiré patterns in HD and UHD images are more complicated because of the severe interference between the much finer patterns, such as the individual pixels on an HD display and the sampling grid of image sensors [7]. In addition to the complexity of moiré patterns in HD and UHD images, an HD/UHD demoiréing method also needs to consider the large amount of memory cost with the increasing image size. FHDe²Net [7] is a pioneer work for the removal of moiré patterns in HD images, in which the moiré patterns are first erased at the low-resolution stage, and the textures are then refined at the high-resolution stage. However, FHDe²Net generates artifacts when performing demoiréing for UHD images, which are more susceptible to moiré patterns due to the higher spatial frequency. In contrast, ESDNet [46] has been proposed recently for the demoiréing in UHD images with promising performance, where a semantic-aligned scale-aware module is designed for the scale variation of moiré patterns. Despite the high performance, two limitations exist in ESDNet: (i) it neglects the internal moiré-specific properties, leading to sub-optimal results with unrealistic artifacts and incorrect tones (see Figure 1 (left)); (ii) when attempting to

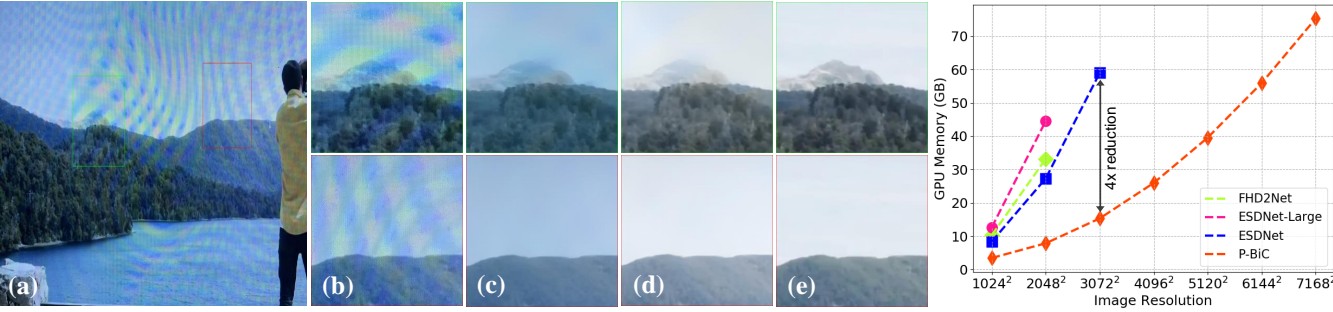

**Figure 1: Left: Examples of ultra-high-definition image demoiréing. (a)-(b) The degraded image with a resolution of** $3840 \times 2160$ **from the UHDM dataset [46]. (c) Results generated by ESDNet-L [46]. (d) Results generated by our proposed P-BiC. (e) Ground-truth images. The presented results effectively highlight the superior capability of P-BiC in eliminating undesired moiré patterns and preserving intricate textures. Right: Comparison of GPU memory utilization among distinct image demoiréing techniques. Missing data points indicate that moiré images at these resolutions cannot be reconstructed on an 80GB GPU (NVIDIA A100) due to memory limitations. P-BiC stands out by not only restoring moiré images up to a resolution of** $7168 \times 7168$**, but also achieving a remarkable 4-fold reduction in memory utilization when restoring an image of** $3072 \times 3072$**, as compared to the method presented in [46]. Notably, these values are derived from feeding the entire image into the GPU for demoiréing.**

directly input the entire image without cropping into the network, the issue of "out of memory" arises, particularly when processing images of higher resolutions. For instance, on a high-performance GPU like the NVIDIA A100 with 80GB of memory, the ESDNet method can handle a maximum resolution of $3072 \times 3072$ before encountering this limitation.

To address the above problems, in this paper, we introduce a novel method named the Patch Bilateral Compensation Network (P-BiC) for UHD image demoiréing. This novel method offers memory efficiency and leverages prior knowledge to guide the process. The key to memory efficiency lies in partitioning a full-resolution feature map into several smaller patches, a technique that enables processing UHD moiré images within the constraints of limited memory resources. To illustrate, P-BiC can conduct demoiréing on a $7168 \times 7168$ resolution image while staying within the memory limits of a single NVIDIA A100 GPU, as depicted in Figure 1 (right).

Furthermore, we leverage a fundamental observation as the guiding principle for our bilateral compensation approach. Specifically, we exploit the fact that the green channel of an image is less affected by moiré patterns, which can be attributed to the red and blue channels having half the sampling frequency of the green channel in the color filter array [1, 6]. This inherent property allows the moiré image to retain well-preserved color information despite the presence of moiré artifacts. Building upon this observation, we disentangle the UHD demoiréing task into two distinct sub-tasks, which are facilitated by the core components of P-BiC: the Green-Guided Detail Transfer (G$^2$DT) module and the Style-Aware Tone Adjustment (STA) module. The G$^2$DT module effectively employs the features of the green channel to complement distorted features with intact content information. Conversely, the STA module undertakes color representation adjustments for the features of the green channels, which might otherwise lose significant tone details. The synergistic integration of these components empowers P-BiC to deliver both efficient GPU memory usage and high-quality demoiréing outcomes, as showcased in Figure 1. These achievements stem from the design

of a patch-level processing strategy and the strategic utilization of prior observations to guide bilateral compensation.

We summarize our contributions as follows. (1) We identify the limitations of existing UHD image demoiréing methods, *i.e.*, the intense memory cost, and the poor-quality results. Inspired by that, a novel network is specifically designed in this work for UHD image demoiréing, achieving both memory efficiency and higher quality results. (2) We propose a novel patch bilateral compensation network (P-BiC) that operates at the patch level to restore UHD moiré images with limited memory use and utilizes a prior observation for bilateral compensation. In particular, two key modules of P-BiC, namely the G$^2$DT module and the STA module, serve for the bilateral compensation. (3) We validate the effectiveness of P-BiC through comprehensive experiments on benchmark datasets. It significantly outperforms existing methods while maintaining lower computational costs, thus highlighting the efficacy and practicality of our proposed method.

## 2 Related Work

**Conventional image demoiréing.** To suppress moiré patterns, Kim *et al.* [9] place an optical low-pass filter in front of the lens to avoid aliasing. However, it cannot eliminate moiré artifacts while reserving the image details. Similarly, it is time-consuming to capture a moiré-free image by selecting an optimal angle of lens [10, 22]. Later methods have relied on various filtering or image decomposition techniques. Wei *et al.* [33] propose a median-Gaussian filtering method for eliminating moiré patterns in X-ray microscopy images. Liu *et al.* [13] utilize a low-rank and sparse matrix decomposition-based method to remove moiré patterns from texture images. Yang *et al.* [42] utilize layer decomposition on polyphase components for demoiréing.

**Deep image demoiréing.** Sun *et al.* [26] propose a multi-scale demoiréing network with the first benchmark dataset that captures real LCD screens for training and evaluating demoiréing models. He *et al.* [6] add annotations of different types of moiré

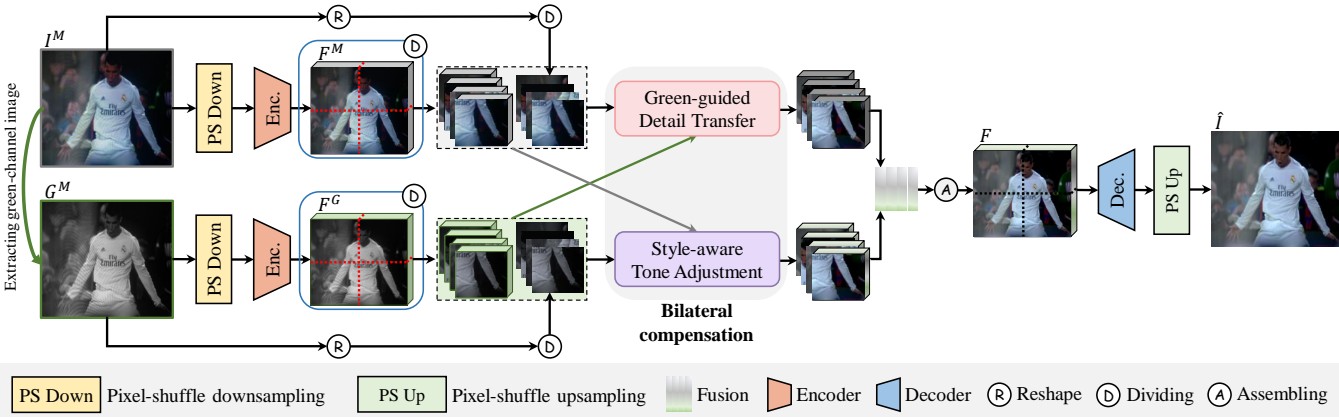

**Figure 2: Overview of our proposed P-BiC. P-BiC can generate a moiré-free image $\hat{I}$ from the moiré-degraded observation $I^M$. After extracting green-channel image $G^M$ from $I^M$, both of them are converted into the feature domain, denoted as $F^M$ and $F^G$. These rescaled features are then divided into patches, followed by the bilateral compensation, which consists of the G²DT module and the STA module. The outputs of both modules are then fused and assembled to $F$. Finally, $F$ is fed to the feature decoder for moiré-free image reconstruction.**

patterns to the dataset in [26] and propose MopNet for demoiréing. Yue *et al.* [48] propose a multiplicative operation-based network that simultaneously removes moiré patterns and improves image brightness. Zheng *et al.* [52, 53] and He *et al.* [7] further exploit DCT domain priors for demoiréing, while Liu *et al.* [16] utilize wavelet domain features to separate the frequencies of moiré patterns. Liu *et al.* [17] introduce an additional input of a focused-defocused image pair for demoiréing with a self-supervision scheme. Wang *et al.* [29] propose a coarse-to-fine disentangling framework for demoiréing. Zhang *et al.* [51] propose a patch-based framework for efficient demoiréing based on existing methods. Recently, Yu *et al.* [46] propose ESDNet, achieving promising performance on image demoiréing.

**High-definition image restoration.** With the rapid advancement of mobile devices, modern smartphones are now capable of capturing HD and UHD images, underscoring the importance of research in HD and UHD image restoration for practical applications. For instance, Zheng *et al.* [54] introduce a dehazing method tailored for UHD images, employing a multi-guided bilateral learning framework. This approach integrates both global and local guidance information to produce more accurate and visually appealing dehazed images. Deng *et al.* [4] propose a multi-scale separable network designed for UHD video deblurring, harnessing both spatial and temporal cues to generate sharp and clear video frames. Yi *et al.* [43] introduce a contextual residual aggregation mechanism by learning the change of image resolution for UHD image inpainting. Feng *et al.* [5] propose GLSGN for UHD image restoration. The GLSGN adopts both local and global pathways to restore images in a step-wise manner, and is effective in deraining, dehazing, and reflection removal. In the realm of image demoiréing, FHDe²Net [7] and ESDNet [46] have been developed to address moiré patterns in HD and UHD images.

**Green channel prior.** As discussed in [28], the CMOS sensor has different sensitivity to light of different wavelengths or colors, and

in most illumination conditions, green channels are brighter than red and blue channels in Bayer pattern CFA images. As a result, the green channel has more texture information than red/blue channels in most natural images. While previous works, such as MopNet [6], have acknowledged that the green channel is less affected by moiré interference, they have not fully exploited this inherent characteristic. Some existing methods [8, 15, 27, 49] utilize the green channel to reconstruct the other color channels. However, they do not delve into the specific details of the green channel or explore its complementary nature with RGB images. In contrast to these approaches, our motivation stems from the observation that the green channel is less impacted by moiré but lacks color information. Additionally, the tone information in moiré images remains largely intact despite color shifts.

## 3 Method

### 3.1 Overview

As shown in Figure 2, given a moiré-degraded image $I^M \in \mathbb{R}^{3 \times H \times W}$, P-BiC can generate a moiré-free image $\hat{I}$, which should be close to the ground-truth image $I^{GT}$. To fully exploit the internal moiré-specific property, we extract the green-channel moiré image $G^M$ from $I^M$, which is less affected by moiré patterns. We first feed $I^M$ and $G^M$ to pre-processing heads to enlarge the receptive field [46], followed by the feature encoders to extract multi-scale features $\{F_1^M, F_2^M, F_3^M\}$ and $\{F_1^G, F_2^G, F_3^G\}$. The sizes of $F_3^M$ and $F_3^G$ are $c \times h \times w$. The encoders consist of three building blocks (convolutional layers and residual blocks), and the second and third blocks halve the size of the feature maps with stride 2. Both encoders do not share weights. To save the memory costs, we divide $F_3^M$ and $F_3^G$ into sequences of small feature patches $\{F_{3,i}^M | i = 1, 2, \ldots, N\}$ and $\{F_{3,i}^G | i = 1, 2, \ldots, N\}$, and we also divide the rescaled images $I^{M\downarrow}$ and $G^{M\downarrow}$ into sequences of small image patches $\{I_i^{M\downarrow} | i = 1, 2, \ldots, N\}$ and $\{G_i^{M\downarrow} | i = 1, 2, \ldots, N\}$. We then feed $\{F_{3,i}^M, F_{3,i}^G, I_i^{M\downarrow}, I_i^{G\downarrow}\}$ to the

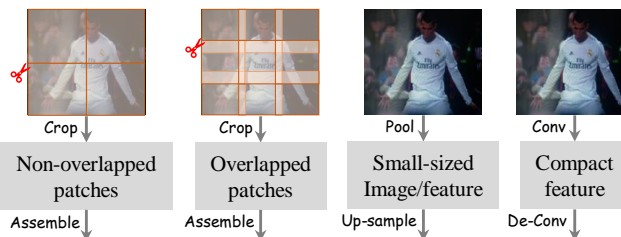

| Crop | Crop | Pool | Conv |
|---|---|---|---|
| Non-overlapped patches | Overlapped patches | Small-sized Image/feature | Compact feature |
| Assemble | Assemble | Up-sample | De-Conv |

**Figure 3: Four patch-dividing strategies toward memory-efficient image demoiréing. See Section 3.2 for details.**

$G^2DT$ module and the STA module for bilateral compensation. Next, the output feature patches from the $G^2DT$ module and the STA module are fused and assembled into a high-resolution feature $F$, which shares the same size as $F_3^M$ and $F_3^G$. Finally, $F$ is fed to the feature decoder, followed by the pixel-shuffle upsampling operation, to obtain the reconstructed moiré-free image $\hat{I}$. The feature encoders and the feature decoder are connected via skip-connections, allowing features containing rich high-resolution information to facilitate the reconstruction of moiré-free images.

## 3.2 Memory-Efficient UHD Image Demoiréing

Image demoiréing methods face significant challenges when dealing with UHD images due to high memory requirements, which can limit their practicality and scalability. We propose a memory-efficient strategy for UHD image demoiréing that centers on reducing the resolution of the processed feature $F$. To this end, we present four possible methods, each with multiple design choices, for achieving memory saving in UHD demoiréing.

**Cropping non-overlapped patches.** Given a high-resolution feature, the reflect padding operation is first adopted, and we then crop $F$ into multiple non-overlapping patches $\{F_i | i = 1, 2, \ldots, N_{no}\}$. Each divided patch share the same size $F_i \in \mathbb{R}^{c \times (h/\sqrt{N_{no}}) \times (W/\sqrt{N_{no}})}$. After we process each divided patch separately with the bilateral compensation, we combine them in their original order.

**Cropping overlapped patches.** We first utilize the reflect padding operation, and we then crop $F$ into multiple overlapped patches $\{F_i | i = 1, 2, \ldots, N_o\}$ using a $K \times K$ pixels sliding window with a stride of $S$. After the loop of patch-level bilateral compensation, we discard the overlapping regions on the processed patches and assemble them into a complete feature.

**Unlearnable rescaling.** We feed a pooled small-sized feature instead of feeding the full-resolution one to the bilateral compensation part. We then upsample the processed feature to the original size.

**Learnable rescaling.** We utilize convolutional layers with different strides for learnable rescaling, and we feed the compact feature to the bilateral compensation stage.

Further experiments demonstrate that cropping non-overlapped patches with $N_{no} = 4$ is an effective strategy that can achieve a favorable balance between performance and computational costs.

## 3.3 Bilateral Compensation

The presence of moiré patterns introduces a challenging task of demoiréing, as they exhibit a wide-ranging frequency spectrum

that intertwines with the underlying images. This complexity is further amplified when dealing with UHD images due to heightened interference. Consequently, it becomes imperative to fully harness the intrinsic moiré-specific characteristics to achieve effective UHD demoiréing. A fundamental insight emerges from the structure of a typical Bayer color filter array: the green channel's sampling frequency is twice that of the red and blue channels. This property results in the green channel being less affected by moiré patterns, with the tone information in the moiré image remaining relatively intact. Our proposed P-BiC capitalizes on this observation by introducing the $G^2DT$ and STA modules. These modules operate at the patch level and engage in bilateral compensation to exploit the moiré-specific properties, ultimately contributing to improved UHD demoiréing results.

**Green-guided detail transfer.** As shown in Figure 4, the $G^2DT$ module is designed to transfer the details contained in green-channel images to the moiré images. We first concatenate the moiré image feature $F_{3,i}^M$ and the moiré feature $I_{3,i}^{M\downarrow}$, followed by a convolutional layer to generate the enhanced moiré feature $F_i^M$. The enhanced green image feature $F_{3,i}^G$ can be obtained in the same way

$$F_{3,i}^M = \text{Conv}([F_{3,i}^M, I_i^{M\downarrow}]), F_{3,i}^G = \text{Conv}([F_{3,i}^G, I_i^{G\downarrow}]), \quad (1)$$

where $[\cdot, \cdot]$ denotes the concatenate operation. Such a process combines the complementary advantages of the feature domain and the image domain, which promote the enrichment of feature representation [25]. $F_{3,i}^M$ and $F_{3,i}^G$ are then fed to the simple yet effective green-channel branch and the moiré branch. The green-channel branch aims to refine and enhance the details of the green image feature. It first convolves the green channel feature $F_{3,i}^G$, and the residual $R_{3,i}^G$ between the convolved green-channel feature and the moiré feature $F_i^M$ is then convolved as

$$\begin{aligned} R_{3,i}^G &= \text{Conv}(F_{3,i}^G) - F_{3,i}^M, \\ F_{3,i}^{G'} &= F_{3,i}^G + \text{Conv}(R_{3,i}^G). \end{aligned} \quad (2)$$

Similarly, the details in the moiré image feature are refined and enhanced as

$$\begin{aligned} R_{3,i}^M &= F_{3,i}^M - \text{Conv}(F_{3,i}^M), \\ F_{3,i}^{M'} &= \text{Conv}(R_{3,i}^M + F_{3,i}^G). \end{aligned} \quad (3)$$

The outputs of both branches are then concatenated and aggregated, obtaining $F_{3,i}'$. In practice, however, we find feed $F_{3,i}'$ to the following part in P-BiC tends to generate artifacts since the $G^2DT$ module is performed at the patch level, ignoring global information contained in high-resolution input. We therefore feed the rescaled green-channel image $I^{G\downarrow}$ with $F_{3,i}'$ to the fusion operation for further aggregation. Please refer to the supplementary for the details of the fusion operation.

**Style-aware tone adjustment.** The green-channel image and the moiré image share the same content and texture, but their color distributions differ. To address this, we introduce the STA module, which aims to map the distribution of the green-channel image feature to that of the moiré image feature for tone adjustment.

As shown in Figure 5, we first obtain the enhanced green image feature $F_{3,i}^G$ and moiré feature $F_{3,i}^M$, which is similar to what we have

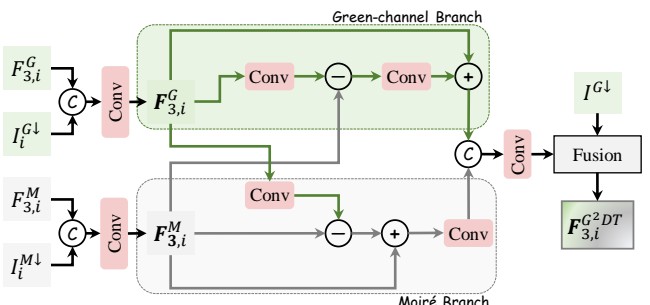

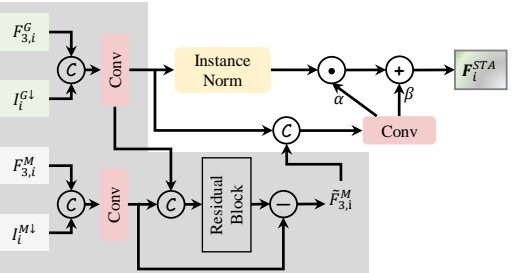

**Figure 5: Structure of the STA module.**

**Figure 4: Left: Structure of the G$^2$DT module. Right: Architecture diagram of the fusion process.**

done in the G$^2$DT module. We then obtain the style feature $\tilde{F}_{3,i}^M$ via

$$\tilde{F}_{3,i}^M = \text{ResB}([F_{3,i}^G, F_{3,i}^M]) - F_{3,i}^M, \tag{4}$$

where $\text{ResB}(\cdot)$ denotes the residual block. The green channel feature and the style feature are initially concatenated before being fed into a convolutional layer, resulting in the generation of two parameters, denoted as $\alpha$ and $\beta$. Both $\alpha$ and $\beta$ possess the same dimensions as the style feature. Then, instance normalization is applied to the green-channel feature as

$$F_{3,i}^{G,c} \leftarrow \frac{(F_{3,i}^{G,c} - \mu_{3,i}^{G,c})}{\sigma_{3,i}^{G,c}}, \tag{5}$$

where $\mu_{3,i}^{G,c}$ and $\sigma_{3,i}^{G,c}$ are the mean and standard deviation of $F_{3,i}^G$ in channel $c$. We then update $\alpha$ and $\beta$ with the mean and standard deviation of the style feature as

$$\alpha \leftarrow \alpha + \mu_{\tilde{F}_{3,i}^M}, \beta \leftarrow \beta + \sigma_{\tilde{F}_{3,i}^M}. \tag{6}$$

Finally, $\alpha$ and $\beta$ are multiplied and added to the normalized green channel feature in an element-wise manner as

$$F_{3,i}^{STA} = \alpha \cdot F_{3,i}^{G,c} + \beta. \tag{7}$$

This process ensures a style-aware adjustment of the tone of the green channel feature.

The STA module draws inspiration from [23]. However, it differentiates itself from [23]. While [23] utilizes segmentation maps to generate two parameters, the convolutions within our STA module accept both the green channel image/feature and the style feature as inputs, enabling them to learn the differences between them. Furthermore, after obtaining the parameters $\beta$ and $\gamma$ from these convolutions, we combine them with the mean and standard deviation of the green channel feature. Such design contributes to the style-aware adjustment of the green channel feature.

Upon obtaining the processed patch features, $F_{3,i}^{G^2DT}$ and $F_{3,i}^{STA}$ are combined through concatenation, followed by a convolutional layer. These aggregated features are then fed into the feature decoder for further reconstruction. It is noteworthy that, akin to ESDNet [46], P-BiC generates three hierarchical moiré-free predictions $\hat{I}3, \hat{I}2, \hat{I}$, with $\hat{I}3$ and $\hat{I}2$ being employed and supervised during the training phase.

### 3.4 Loss Functions

**Reconstruction loss.** We adopt $L_1$ loss as the reconstruction loss to supervise hierarchical moiré-free predictions as

$$\mathcal{L}_{rec} = ||I^{GT} - \hat{I}||_1 + ||I_i^{GT} - \hat{I}_i||_1, i = 2, 3, \tag{8}$$

where $I_i^{GT}$ is the rescaled ground-truth image, which has the same size as $\hat{I}_i$.

**Perceptual loss.** The perceptual loss is expressed as

$$\mathcal{L}_{per} = ||\phi_j(I^{GT}) - \phi_j(\hat{I})||_1 + ||\phi_j(I_i^{GT}) - \phi_j(\hat{I}_i)||_1, i = 2, 3, \tag{9}$$

where $\phi_j(\cdot)$ denotes the $j$-th layer of the pretrained VGG16 network. Here, we use conv3_3 (after ReLU).

**Full objective.** Our full objective is defined as

$$\mathcal{L} = \mathcal{L}_{rec} + \lambda \mathcal{L}_{per}, \tag{10}$$

where $\lambda$ is the weighting factor to balance two loss terms.

## 4 Experiments

### 4.1 Experimental Setting

**Datasets.** We conduct experiments on four public image demoiréing datasets: TIP2018 [26], LCDMoiré [47], FHDMi [7], and UHDM [46]. The TIP2018 dataset comprises 150,000 real image pairs, with 135,000 images used for training and the remaining images for testing. The dataset is constructed by capturing photographs of the ImageNet dataset displayed on computer screens with different hardware configurations. The LCDMoiré dataset comprises 10,200 image pairs that are synthetically generated, consisting of 10,000 images for training and 100 images each for validation and testing. The FHDMi dataset contains 9,981 image pairs for training and 2,019 for testing with the resolution of $1920 \times 1080$ for HD image demoiréing. UHDM is a new benchmark dataset, with 5,000 image pairs specifically designed for UHD image demoiréing. It features diverse moiré patterns commonly found in UHD images.

**Training and testing settings.** For the TIP2018 dataset, we initially resize the images to a resolution of $286 \times 286$, followed by a central cropping step to generate $256 \times 256$ resolution images for both training and testing purposes. Concerning the FHDMi and LCDmoiré datasets, we perform random cropping of $512 \times 512$ patches from HD images for training, while maintaining the original resolution images for testing. Regarding the UHDM dataset, our training of P-BiC utilizes cropped patches. For testing, we perform center cropping on the original images to generate test pairs with a resolution of $3840 \times 2160$, consistent with [46].

**Table 1: Quantitative comparisons in terms of PSNR, SSIM, and LPIPS between P-BiC and state-of-the-art demoiréing methods on benchmark datasets. The best results are marked in bold while the second ones are marked with underlines.**

| Dataset | Metrics | Input | DMCNN [26] | MDDM [2] | WDNet [16] | MopNet [6] | MBCNN [52] | FHDe²Net [7] | ESDNet [46] | ESDNet-L [46] | Wang [29] | P-BiC |
|---|---|---|---|---|---|---|---|---|---|---|---|---|
| TIP2018 | PSNR↑ | 20.30 | 26.77 | - | 28.08 | 27.75 | 30.03 | 27.78 | 29.81 | 30.11 | 28.87 | **30.56** |
| | SSIM↑ | 0.7380 | 0.8710 | - | 0.9040 | 0.8950 | 0.8930 | 0.8960 | 0.9160 | 0.9200 | 0.9840 | **0.9250** |
| LCDMoiré | PSNR↑ | 10.44 | 35.48 | 42.49 | 29.66 | - | 44.04 | 41.40 | 44.83 | 45.34 | - | **45.55** |
| | SSIM↑ | 0.5717 | 0.9785 | 0.9940 | 0.9670 | - | 0.9948 | - | 0.9963 | 0.9966 | - | **0.9972** |
| FHDMi | PSNR↑ | 17.97 | 21.54 | 20.83 | - | 22.76 | 22.31 | 22.93 | 24.50 | 24.88 | - | **25.45** |
| | SSIM↑ | 0.7033 | 0.7727 | 0.7343 | - | 0.7958 | 0.8095 | 0.7885 | 0.8351 | 0.8440 | - | **0.8473** |
| | LPIPS↓ | 0.2837 | 0.2477 | 0.2515 | - | 0.1794 | 0.1980 | 0.1688 | 0.1354 | **0.1301** | - | 0.1493 |
| UHDM | PSNR↑ | 17.12 | 19.91 | 20.09 | 20.36 | 19.49 | 21.41 | 20.34 | 22.12 | 22.42 | - | **23.30** |
| | SSIM↑ | 0.5089 | 0.7575 | 0.7441 | 0.6497 | 0.7572 | 0.7932 | 0.7496 | 0.7956 | 0.7985 | - | **0.8007** |
| | LPIPS↓ | 0.5314 | 0.3764 | 0.3409 | 0.4882 | 0.3857 | 0.3318 | 0.3519 | 0.2551 | 0.2454 | - | **0.2324** |
| - | Params (M) | - | **1.426** | 7.637 | 3.360 | 58.565 | 14.192 | 13.571 | 5.934 | 10.623 | 15.400 | 4.922 |

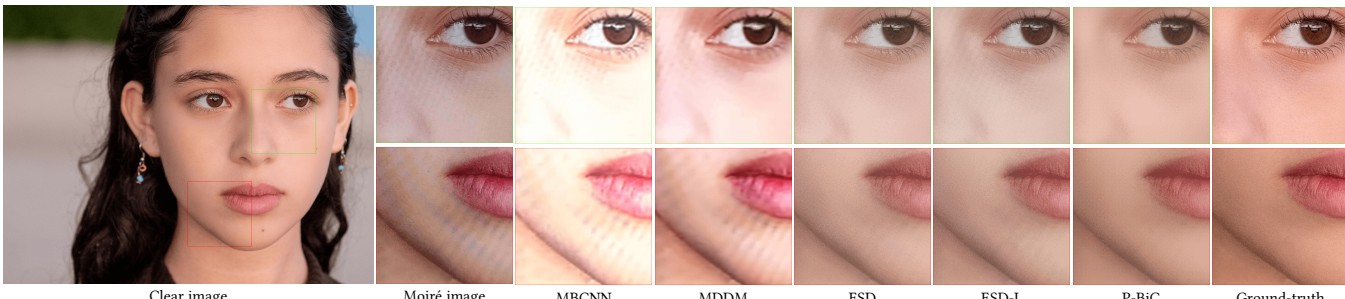

Clear image — Moiré image — MBCNN — MDDM — ESD — ESD-L — P-BiC — Ground-truth

**Figure 6: Qualitative comparisons of different image demoiréing methods on the UHDM dataset.**

**Evaluation metrics.** For our evaluation, we employ widely-accepted image quality assessment metrics, including PSNR, SSIM [32], and LPIPS [50]. Past studies have indicated that LPIPS offers greater consistency with human perception, making it particularly suitable for assessing demoiréing results [7, 46]. It is crucial to emphasize that in line with established conventions in the field, existing methods have typically used only PSNR and SSIM metrics on the TIP2018 and LCDmoiré datasets. In order to ensure fairness and comparability, we follow this convention in our evaluations.

**Implementation details.** We implement our algorithm using PyTorch on a single NVIDIA RTX 3090 GPU. We utilize the Adam optimizer with parameters $\beta_1 = 0.9$ and $\beta_2 = 0.999$. The learning rate is initially set to 0.0002 and scheduled by cyclic cosine annealing. We set $\lambda = 1$ to balance different loss terms. Our P-BiC is trained with different configurations depending on different datasets. For TIP2018, we train for 100 epochs with a batch size of 4. For FHDMi and LCDMoiré, we train for 200 epochs with a batch size of 2. Similarly, for UHDM, we train for 200 epochs with a batch size of 2.

## 4.2 Quantitative and Qualitative Comparisons

We conduct a comprehensive comparison between our P-BiC and several state-of-the-art image demoiréing methods that have publicly available source code. These methods include DMCNN [26], MDDM [2], WDNet [16], MopNet [6], MBCNN [52], FHDe²Net [7], Wang [29], ESDNet, and its larger variant ESDNet-L [46]. It is worth noting that DDA [51] is designed specifically for low-resolution moiré images and aims to enhance the performance and efficiency

**Table 2: Computational cost comparisons measured on an NVIDIA A100 GPU between P-BiC and UHD image demoiréing methods on UHDM. In our evaluations, we feed the entire UHD moiré image as the input. The PSNR results of ESDNet and ESDNet-L are extracted from [46]. The unit of runtime is second, and memory here denotes the peak memory (GB).**

| Method | Runtime | Memory | PSNR |
|---|---|---|---|
| ESDNet | 0.140 | 24.58 | 22.12 |
| ESDNet-L | - | - | 22.95 |
| P-BiC | 0.268 | 19.77 | 23.21 |

of existing networks. Therefore, we do not include it in the comparison, as its design philosophy is not directly applicable to the UHD image demoiréing task addressed in this paper.

**Quantitative comparison.** Table 1 provides a quantitative comparison across four benchmark testsets, highlighting the consistently superior performance of P-BiC. Notably, P-BiC demonstrates significant outperformance compared to the state-of-the-art UHD image demoiréing method, ESDNet-L. Specifically, P-BiC achieves improvements of 0.45dB, 0.21dB, 0.57dB, and 0.88dB in terms of PSNR on the TIP2018, LCDmoiré, FHDMi, and UHDM datasets, respectively. An important observation is that P-BiC attains these remarkable results while utilizing only 4.922M parameters, in stark contrast to ESDNet-L's utilization of 10.623M parameters. This emphasizes P-BiC's efficiency in UHD image demoiréing.

**Table 3: Comparisons of different methods towards memory-efficient UHD image demoiréing.**

| Method | Detail | Runtime | Memory | PSNR |
|---|---|---|---|---|
| Non-overlap | $N_{no} = 4$ | 0.268 | 19.77 | 23.21 |
| | $N_{no} = 16$ | 0.361 | 20.05 | 23.22 |
| | $N_{no} = 64$ | 0.575 | 19.81 | 23.17 |
| | $N_{no} = 256$ | 1.773 | 19.73 | 23.06 |
| Overlap | $K = 16, S = 8$ | 1.218 | 21.97 | 22.22 |
| | $K = 32, S = 8$ | 2.568 | 51.15 | 22.58 |
| | $K = 64, S = 16$ | 1.783 | 44.80 | 22.77 |
| Unlearnable | Maxpool=2 | 0.145 | 11.88 | 21.47 |
| | Maxpool=4 | 0.131 | 11.31 | 20.05 |
| | Maxpool=8 | 0.129 | 11.17 | 17.29 |
| | Avgpool=2 | 0.136 | 11.83 | 19.92 |
| | Avgpool=4 | 0.134 | 11.29 | 19.47 |
| | Avgpool=8 | 0.130 | 11.17 | 13.23 |
| Learnable | Stride=2 | 0.142 | 11.83 | 21.65 |
| | Stride=4 | 0.105 | 11.29 | 21.23 |
| | Stride=8 | 0.131 | 11.17 | 21.27 |
| | Stride=16 | 0.129 | 11.04 | 21.16 |

**Table 4: Analysis on the bilateral compensation in P-BiC.**

| Method | TIP2018 | FHDMi | UHDM |
|---|---|---|---|
| P-BiC-w/o-$G^2DT$ | 30.18 | 25.08 | 23.04 |
| P-BiC-w/o-STA | 30.06 | 24.85 | 22.95 |
| P-BiC-w/o-fusion | 30.49 | 25.40 | 23.21 |
| P-BiC | 30.56 | 25.45 | 23.30 |

**Table 5: Analysis on the $G^2DT$ module in P-BiC.**

| Method | TIP2018 | FHDMi | UHDM |
|---|---|---|---|
| $G^2DT$-Cat | 30.21 | 25.18 | 23.11 |
| $G^2DT$-Resblock | 30.36 | 25.25 | 23.22 |
| $G^2DT$-w/o-green | 30.40 | 25.33 | 23.23 |
| $G^2DT$-w/o-moiré | 30.46 | 25.32 | 23.19 |
| $G^2DT$ | 30.56 | 25.45 | 23.30 |

**Table 6: Analysis on the STA module in P-BiC.**

| Method | TIP2018 | FHDMi | UHDM |
|---|---|---|---|
| STA-Cat | 30.16 | 25.06 | 23.01 |
| STA-Resblock | 30.18 | 25.17 | 23.17 |
| STA-w/o-Adjustment | 30.20 | 25.23 | 23.21 |
| STA | 30.56 | 25.45 | 23.30 |

The computational costs are detailed in Table 2. It is worth noting that, for our comparison, we input the entire UHD moiré image into the network for inference. While existing methods can employ patch-wise inference followed by the merging operation, this approach often leads to artifacts at the borders and suboptimal results. Additionally, border pixels may not fully benefit from neighboring pixels outside the patch for image restoration [3]. To address these issues, we perform a comprehensive evaluation by conducting inference on the entire image. Conclusively, P-BiC distinctly surpasses ESDNet in terms of both performance and computational efficiency. Another illustration of computational efficiency can be observed in Figure 1. It is evident that our P-BiC is capable of operating at higher resolutions, whereas ESDNet and ESDNet-L struggle in this aspect.

**Qualitative comparison.** Exemplar visual results for various methods are presented in Figure 1 and Figure 6. In Figure 1, it is notable that only P-BiC is able to generate color-accurate results, whereas other methods fail to reconstruct the colors accurately. Shifting our focus to Figure 6, the comparison becomes even more compelling. In this context, it becomes exceedingly clear that P-BiC consistently generates results of a superior nature, characterized by enhanced details and precise color rendering. This, in turn, contributes to the generation of more perceptually pleasing and visually captivating moiré-free results.

### 4.3 Further Analysis

**Memory-efficient UHD image demoiréing.** Table 3 presents a comprehensive comparison of four distinct patch-dividing strategies, all geared towards achieving memory-efficient UHD demoiréing. These strategies encompass non-overlapping patch cropping with varying quantities, overlapping patch cropping characterized by different window sizes ($K$) and strides ($S$), unlearnable methodologies grounded in maxpooling and avgpooling operations, and the learnable approach which relies on convolutional layers with diverse strides. It is evident from the results that the unlearnable strategy yields the least favorable reconstruction outcomes. This is attributed

to the inherent limitations of utilizing pooling operations for direct resolution reduction, which results in irreversible information loss and consequently, suboptimal results. In contrast, the strategy centered around cropping patches exhibits improved performance. Additionally, a trade-off between the number of non-overlapping patches and reconstruction performance becomes apparent. While increasing the count of cropped patches aids in curbing memory usage, it simultaneously elongates runtime and dampens PSNR values. This stems from the fact that an upsurge in the number of cropped patches leads to a decrease in the resolution of individual patches, thereby constraining the extent of global information available for moiré pattern removal. Given these findings, we opt for the non-overlapping strategy, setting the number of cropped patches at 4. This choice ensures a judicious balance between performance and computational efficiency, corroborated by empirical evidence.

**Effectiveness of bilateral compensation.** Bilateral compensation, leveraging the intrinsic moiré-specific characteristics, is a key component of our approach. To showcase its effectiveness, we design and analyze several variants: (1) BiC-w/o-$G^2DT$: in this variant, we directly remove the $G^2DT$ module. (2) BiC-w/o-STA: this variant involves the direct removal of the STA module. (3) BiC-w/o-fusion: we replace the fusion operation with a simple addition operation. When removing the $G^2DT$ module and STA module, we utilize residual blocks to maintain the same parameters. We present the quantitative results of these variants in Table 4. Notably, upon removing the $G^2DT$ module and the STA module, the PSNR values on the UHDM dataset experience reductions of 0.26dB and 0.35dB, respectively. This indicates the pivotal role played by these two modules in the bilateral compensation process. Additionally,

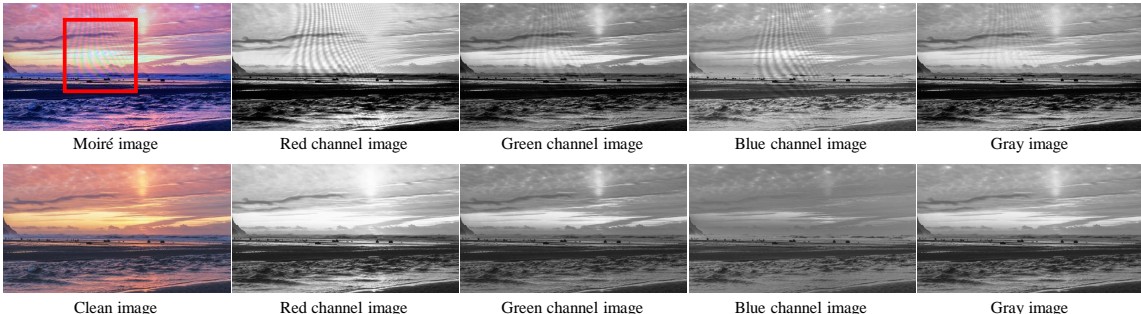

**Figure 7: Comparison of red, green, and blue channel images along with grayscale images for moiré and clear images.**

upon incorporating the fusion operation, we observe a 0.09dB PSNR enhancement on the UHDM dataset.

**Effectiveness of the G$^2$DT module.** We design the following variants to demonstrate the effectiveness of the G$^2$DT module. (1) G$^2$DT-Cat: we concatenate all inputs directly and feed them to the following processes. (2) G$^2$DT-Resblock: we feed all the inputs to several residual blocks. (3) G$^2$DT-w/o-green: we remove the green-channel branch. (4) G$^2$DT-w/o-moiré: we remove the moiré branch. Quantitative results are presented in Table 5. As is evident, the G$^2$DT module outperforms G$^2$DT-Cat and G$^2$DT-Resblock by 0.19dB and 0.08dB, respectively. Moreover, removing either branch leads to a decline in performance, highlighting the crucial role of our dual-branch design.

**Effectiveness of the STA module.** To effectively showcase the significance of the STA module, we have designed the following variants: (1) STA-Cat: in this variant, all inputs are directly concatenated and then fed into subsequent processes. (2) STA-Resblock: all inputs are passed through several residual blocks in this variant. (3) STA-w/o-Adjustment: here, we remove the style-aware tone adjustment operation from the original STA module. The quantitative results are presented in Table 6. Notably, a significant performance drop is observed when the tone adjustment operation is removed. Specifically, on TIP2018, FHDMi, and UHDM datasets, the performance decreases by 0.36dB, 0.22dB, and 0.09 dB, underlining the critical importance of the tone adjustment operation.

**Effectiveness of the green channel.** As analyzed earlier, the green channel contains crucial information that can be effectively utilized for UHD demoiréing. To visually illustrate this, we visualize the red, green, blue, and grayscale versions of both moiré-distorted and clear images in Figure 7. The visualization indeed confirms that the green channel is less impacted by moiré patterns compared to the other channels.

### 4.4 Limitations and Discussions

Despite the promising performance showcased above, P-BiC does have certain limitations that warrant further investigation. In real-world scenarios, especially when image texture and moiré patterns are intricately intertwined, P-BiC might encounter challenges in effectively discriminating between these elements. Please refer to the supplementary material.

In light of these limitations, our future research direction will encompass various areas to enhance P-BiC's capabilities and applicability: (1) Real-time processing:. One crucial avenue for future research involves bridging the gap towards real-time performance. To achieve this, we plan to design more efficient and lightweight architectures tailored for real-time UHD demoiré processing. This will contribute to making P-BiC a more practical solution for applications requiring instant processing. (1) Advanced patch cropping. While our exploration into patch cropping strategies is valuable, future work can delve into more sophisticated techniques, such as irregular patch cropping. These methods can adapt to diverse moiré pattern shapes and distributions, enhancing the versatility of P-BiC across a wider range of scenarios. (3) Extension to other tasks. Beyond UHD demoiréing, we intend to extend the utility of P-BiC to other UHD image reconstruction tasks [11, 30, 31, 45]. (4) Video demoiréing. Given the success of P-BiC in UHD image demoiréing, an exciting avenue involves its expansion to video demoiréing tasks [35–38]. Extending P-BiC's capabilities to handle video sequences with moiréing patterns would further amplify its practical relevance and applicability.

### 5 Conclusion

In this paper, we address the limitations of existing UHD image demoiréing methods by proposing a novel patch bilateral compensation network (P-BiC) that achieves both the memory efficiency and the high-quality UHD image demoiréing. Specifically, P-BiC performs UHD moiré pattern removal at the patch level to maintain the low memory cost, and leverages an internal moiré-specific property to enable bilateral compensation via two key modules, namely the G$^2$DT module and the STA module. The G$^2$DT module supplements moiré-distorted features with the intact content and the STA module performs style-aware tone adjustment for color correction. P-BiC achieves state-of-the-art performance on diverse datasets, surpassing previous methods by significant margins. Moreover, P-BiC achieves these results while maintaining low computational costs, making it a practical solution for UHD moiré pattern removal.

### Acknowledgments

This project is supported by the National Research Foundation, Singapore, under its Medium Sized Center for Advanced Robotics Technology Innovation.

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
