# OpenReview forum: "P-BiC: Ultra-High-Definition Image Demoireing via Patch Bilateral Compensation"
_acmmm.org/ACMMM/2024/Conference — MM2024 Poster_

### Official Review · Reviewer_x4mp · 2024-05-24

**Rating:** 3
**Confidence:** 3

**Summary:**

This paper presents a supervised method for the demoireing task. To enhance memory efficiency, training was conducted using small patches (P-BiC). By utilizing the relatively abundant information in the green channel, the method successfully controls details (by G^2DT) and color tone (by STA), thereby improving reconstruction performance.

**Strengths:**

- The method utilizes the characteristics of the green channel in RAW data (e.g., Bayer CFA) for demoireing and provides a thorough analysis related to this approach.
- In terms of fidelity, the method outperforms existing methods.

**Limitations:**

- Although four types of patch utilization are presented, the main contribution seems to be achieving memory efficiency by applying simple non-overlapped patches without introducing any special patch cropping techniques. Feeding image patches instead of the entire image is a well-known approach for efficiency, so it is unclear how this can be considered a novelty. The improvement in peak memory shown in Table 2 does not seem significant, and the runtime nearly doubles, which is critical for real-time applications. To emphasize efficiency, it would have been better to present other metrics such as total memory usage, FLOPS, and BOPS.
- In Table 1, LPIPS is measured in only two tables. It would have been more reliable for assessing perceptual quality related to color tone if LPIPS had been measured in the other tables as well. Additionally, using FID scores could be another method for evaluation.

**Suitability:**

3

---

### Official Review · Reviewer_7JtT · 2024-05-24

**Rating:** 4
**Confidence:** 4

**Summary:**

This work focuses on addressing the problem of ultra-high-definition image demoireing. Specifically, the authors propose a memory-efficient method called P-BiC for removing moire patterns in ultra-high-definition (UHD) images. Specifically, by dividing UHD images into patches, P-BiC performs patch-level demoireing, reducing memory requirements. In addition, leveraging insights about the green channel's resilience to moire patterns, P-BiC guides detail transfer and tone adjustment for effective bilateral compensation. The experimental results show the effectiveness of the proposed method.

**Strengths:**

1. The paper is overall well-written and easy to follow.
2. The proposed method achieves better performance than existing methods.
3. This work provides extensive ablation study experiments, which have verified the effectiveness of the proposed method.

**Limitations:**

1. The paper lacks a comparison of results on real-life examples for UHD demoireing, which is essential for evaluating the method's performance in practical scenarios.
2. Unclear motivation and lack of insight. The authors claim that the green channel of an image remains relatively less affected by moiré patterns. However, there is no evidence to verify this phenomenon.
3. Some related UHD restoration methods, such as LLFormer [a], UHDFour [b], and NSEN [c], are not discussed in the related work section. [a]. Wang et al., Ultra-high-definition low-light image enhancement: A benchmark and
transformer-based method. [b]. Li et al., Embedding fourier for ultra-high-definition low-light image enhancement. [c]. Yu et al., Learning Non-Uniform-Sampling for Ultra-High-Definition Image Enhancement.

**Suitability:**

3

---

### Official Review · Reviewer_Sono · 2024-05-24

**Rating:** 4
**Confidence:** 3

**Summary:**

This manuscript proposes an image moire removal method for UHD images, namely P-BiC. P-BiC divides the image into smaller patches and processes them individually, thus reducing memory usage. It also leverages the observation that the green channel of an image is less affected by moire patterns than the red and blue channels. This allows P-BiC to use the green channel as a guide to restoring the other channels, effectively removing the moire patterns while preserving image details and color accuracy. The method has been evaluated on benchmark datasets and has shown superior performance compared to existing methods.

**Strengths:**

+ This paper is well-motivated. The demoiring on UHD images does pose a challenge to existing methods.
+ The quantitative and qualitative results show its superiority over existing methods.
+ The manuscript is in good structure and is easy to follow.

**Limitations:**

+ Tab.1, There seems to be a typo in SSIM for Wang et al. on TIP2018
+ Although claimed to handle 7000x7000 images, the authors don't present results under that scenario. I don't consider this as a serious limitation.

**Suitability:**

2

---

### Meta-Review · Area_Chair_nayb · 2024-07-02

**Recommendation:** Accept (Poster)
**Confidence:** 5

**Metareview:**

This submission has been reviewed by three experts, the final ratings from whom are all positive after rebuttal (although one of them held a negative vote in initial phase, after rebuttal, the reviewer raised his/her score). Considering the quality of the submission, the comments from the reviewers, and the rebuttal from the authors, the paper can be accepted by the conference.